# Comparing the Psychometric Properties among Three Versions of the UCLA Loneliness Scale in Individuals with Schizophrenia or Schizoaffective Disorder

**DOI:** 10.3390/ijerph19148443

**Published:** 2022-07-11

**Authors:** Chung-Ying Lin, Ching-Shu Tsai, Cian-Ruei Jian, Shu-Ru Chao, Peng-Wei Wang, Huang-Chi Lin, Mei-Feng Huang, Yi-Chun Yeh, Tai-Ling Liu, Cheng-Sheng Chen, Ya-Ping Lin, Shu-Ying Lee, Ching-Hua Chen, Yun-Chi Wang, Yu-Ping Chang, Yu-Min Chen, Cheng-Fang Yen

**Affiliations:** 1Institute of Allied Health Sciences, College of Medicine, National Cheng Kung University, Tainan 70101, Taiwan; cylin36933@gs.ncku.edu.tw; 2Biostatistics Consulting Center, National Cheng Kung University Hospital, College of Medicine, National Cheng Kung University, Tainan 70101, Taiwan; 3Department of Public Health, College of Medicine, National Cheng Kung University, Tainan 70101, Taiwan; 4Department of Occupational Therapy, College of Medicine, National Cheng Kung University, Tainan 70101, Taiwan; 5Department of Child and Adolescent Psychiatry, Chang Gung Memorial Hospital, Kaohsiung Medical Center, Kaohsiung 83301, Taiwan; jingshu@cgmh.org.tw; 6School of Medicine, Chang Gung University, Taoyuan 33302, Taiwan; 7Department of Psychiatry, Kaohsiung Medical University Hospital, Kaohsiung 80756, Taiwan; lifekeeper97@gmail.com (C.-R.J.); 990107@gap.kmu.edu.tw (P.-W.W.); cochigi@kmu.edu.tw (H.-C.L.); 1010236@kmuh.org.tw (M.-F.H.); y7552156@gmail.com (Y.-C.Y.); dai32155@gmail.com (T.-L.L.); sheng@kmu.edu.tw (C.-S.C.); 860146@kmuh.org.tw (Y.-P.L.); 860159@kmuh.org.tw (S.-Y.L.); 8Department of Psychiatry, School of Medicine and Graduate Institute of Medicine, College of Medicine, Kaohsiung Medical University, Kaohsiung 80708, Taiwan; 9Department of Social Work, National Pingtung University of Science and Technology, Pingtung 91201, Taiwan; shanchao@mail.npust.edu.tw; 10Department of Nursing, Kaohsiung Medical University Hospital, Kaohsiung 80756, Taiwan; 940073@mail.kmuh.org.tw (C.-H.C.); 1070289@mail.kmuh.org.tw (Y.-C.W.); 11School of Nursing, The State University of New York, University at Buffalo, Buffalo, NY 14260, USA; yc73@buffalo.edu; 12College of Professional Studies, National Pingtung University of Science and Technology, Pingtung 91201, Taiwan

**Keywords:** confirmatory factor analysis, loneliness, psychological well-being, psychometric properties, schizoaffective, schizophrenia, UCLA Loneliness Scale

## Abstract

The UCLA Loneliness Scale (Version 3; UCLA-LSV3) is widely used for assessing loneliness. Nevertheless, the validity of this scale for assessing loneliness in individuals with schizophrenia or schizoaffective disorder has not been determined. Additionally, studies validating the eight-item and three-item versions of UCLA-LSV3 have not included individuals with severe mental illness; therefore, whether the short versions are comparable to the full 20-item version of UCLA-LSV3 for this population is unclear. The present study examined the unidimensional structure, internal consistency, concurrent validity, and test–retest reliability of the Chinese versions of UCLA-LSV3 (i.e., 20-item, 8-item, and 3-item versions) to determine which version is most appropriate for assessing loneliness in individuals with schizophrenia or schizoaffective disorder in Taiwan. A total of 300 participants (267 with schizophrenia and 33 with schizoaffective disorder) completed the scales, comprising UCLA-LSV3, the Center for Epidemiological Studies Depression Scale (CES-D), the suicidality module of the Kiddie Schedule for Affective Disorders and Schizophrenia–Epidemiological Version (K-SADS-E), and the family and peer Adaptation, Partnership, Growth, Affection, and Resolve (APGAR) index. Construct validity was evaluated through confirmatory factor analysis. The three versions of UCLA-LSV3 were compared with the CES-D, the suicidality module of the K-SADS-E, and the family and peer APGAR index to establish concurrent validity. The results indicated that all three versions of UCLA-LSV3 exhibited acceptable to satisfactory psychometric properties in terms of unidimensional constructs, concurrent validity, and test–retest reliability. The full version of UCLA-LSV3 had the best performance, followed by the eight-item version and the three-item version. Moreover, the three versions had relatively strong associations with each other. Therefore, when deliberating which version of UCLA-LSV3 is the best choice for assessing loneliness in individuals with schizophrenia or schizoaffective disorder, healthcare providers and therapists should consider time availability and practicality.

## 1. Introduction

Loneliness is a key psychological concept and refers to an individual’s perceived distance between anticipated and actual levels of social connectivity [1]. Loneliness is prevalent among individuals with schizophrenia or schizoaffective disorder [2,3,4] and has been identified as a risk factor for physical health problems [4], depression [2,4], anxiety [2,5], substance abuse [4], low self-esteem [3], pessimism [2], and compromised quality of life [2,6,7]. Accordingly, mental health professionals must be provided with a validated instrument for assessing loneliness in individuals with schizophrenia or schizoaffective disorder.

The UCLA Loneliness Scale (Version 3; UCLA-LSV3) [8] is a potential candidate for assessing loneliness in individuals with schizophrenia or schizoaffective disorder owing to its worldwide popularity and promising psychometric evidence. UCLA-LSV3 has been found to have high validity (i.e., concurrent and construct validity) and reliability (i.e., internal consistency and test–retest reliability) across different ethnicities and various languages, such as Farsi [9], Turkish [10], Japanese [11], Danish [12], English [13,14], Spanish [15], Chinese [16,17], and many others [13].

Despite the validity and reliability of UCLA-LSV3, the psychometric properties of this scale and its other versions warrant attention. First, the Chinese versions of UCLA-LSV3 have never been validated among individuals with schizophrenia or schizoaffective disorder in Taiwan, according to a review of the literature; therefore, further psychometric evidence must be derived for these versions.

Second, different versions of UCLA-LSV3 have been proposed, but studies have not compared their psychometric properties or evidence regarding their suitability for assessing individuals with schizophrenia or schizoaffective disorder. In addition to the full version of UCLA-LSV3, which contains 20 items, several short versions of UCLA-LSV3 have been developed [16,18,19]. Among them, the eight- and three-item versions of UCLA-LSV3 have been recommended because their psychometric properties have been determined to be stronger than those of other short versions [18,20]. The eight- and three-item Chinese versions of UCLA-LSV3 have been tested for their psychometric properties, which have been reported to be satisfactory [16,17]. Nevertheless, the eight- and three-item versions of UCLA-LSV3 have not been tested on individuals with severe mental illnesses; therefore, the psychometric properties of these versions for populations with schizophrenia or schizoaffective disorder are unknown. Schizophrenia or schizoaffective disorder may affect an individual’s cognition [21]; thus, individuals with these disorders are likely to interpret UCLA-LSV3 differently when compared with individuals without said disorders. Accordingly, mental health professionals must be provided with information explaining how different versions of UCLA-LSV3 (the full 20-item version, 8-item version, and 3-item version) perform when applied to individuals with schizophrenia or schizoaffective disorder.

Third, the factor structure of UCLA-LSV3 is still uncertain. By definition, loneliness should be considered a unidimensional concept [22]; however, empirical evidence regarding the psychometric properties of UCLA-LSV3 and other instruments for assessing loneliness reveals that these instruments may be multidimensional [23,24,25,26,27,28,29]. Nevertheless, studies have demonstrated a clear pattern of the unidimensionality of loneliness when controlling for method effects (i.e., positively and negatively worded items) in global versions of UCLA-LSV3, namely the Farsi [9], Turkish [10], and English versions [8]. Accordingly, the 20-, 8-, and 3-item versions of UCLA-LSV3 should all be treated as unidimensional instruments for assessing loneliness, although their factor structures could be influenced by method effects.

On the basis of the preceding literature review, the present study examined the unidimensional structures of three Chinese versions of UCLA-LSV3, namely the full 20-item version, 8-item version, and 3-item version, in individuals with schizophrenia or schizoaffective disorder. Moreover, we tested and compared these three versions in terms of internal consistency, concurrent validity, and test–retest reliability to determine which version is most appropriate for assessing loneliness in individuals with schizophrenia or schizoaffective disorder. We observed that shorter versions of UCLA-LSV3 may reduce the time burden for obtaining loneliness information, whereas longer versions may provide comprehensive information about loneliness. Accordingly, mental health professionals must know how much information could be lost when they use a shorter version and must be able to make informed decisions about whether the 20-item version is worth the additional time investment. Moreover, if the scenarios for the shorter versions of UCLA-LSV3 are comparable to those for the full 20-item version, mental health professionals can simply use the shorter versions to reduce the administrative burden of completing the 20-item version.

## 2. Materials and Methods

### 2.1. Participants and Recruitment Process

The present study used convenience sampling to recruit participants from the psychiatric outpatient units of Kaohsiung Medical University Hospital and two community psychiatric rehabilitative institutes in Kaohsiung, Taiwan, from February 2022 to May 2022. The inclusion criteria were as follows: (1) being aged 20–70 years and (2) receiving a diagnosis of schizophrenia or schizoaffective disorder made on the basis of the diagnostic criteria in the fifth edition of the *Diagnostic and Statistical Manual of Mental Disorders* (*DSM-5*) [30]. The exclusion criteria were as follows: (1) having an intellectual disability and (2) having cognitive dysfunction caused by alcohol, substance abuse, or brain injury. Psychiatrists confirmed the eligibility of 362 individuals and invited them to participate in this study. A total of 300 (82.9%) individuals agreed to participate and provided written informed consent prior to completing the survey. Trained research assistants conducted face-to-face interviews with the participants in the interview rooms of the psychiatric outpatient unit affiliated with Kaohsiung Medical University Hospital. The interviews were conducted to collect the participants’ responses regarding experiences of loneliness, depression, suicidal ideation, and perceived support from family and friends. The participants were assured that their responses would remain confidential. Each interview lasted 25 to 40 min, varying by the participant. The Institutional Review Board of Kaohsiung Medical University Hospital approved the study (KMUHIRB-SV(II)-20210097).

### 2.2. Measures

#### 2.2.1. UCLA-LSV3

UCLA-LSV3 comprises 20 items that are used to assess loneliness, and each item is rated on a scale with anchors ranging from 1 (i.e., “never”) to 4 (i.e., “always”). Nine items were reverse coded to be in the same direction as the rest of the items; a higher total UCLA-LSV3 score indicates a higher level of loneliness [8]. Studies on the 20-, 8-, and 3-item versions of UCLA-LSV3 have demonstrated acceptable to satisfactory psychometric properties [8,16,17,18,19,20]. For example, considering internal consistency, the Cronbach α values derived for the 20-, 8-, and 3-item versions of UCLA-LSV3 have been reported to be 0.89–0.94 [8], 0.84 [18], and 0.72–0.87 [16,19], respectively. All three Chinese versions of UCLA-LSV3 also demonstrated favorable psychometric properties among people without psychotic symptoms [16,17].

#### 2.2.2. Center for Epidemiological Studies Depression Scale

The Center for Epidemiological Studies Depression Scale (CES-D) contains 20 items that are used to assess depression, and all items are rated on a scale with anchors ranging from 0 (“rarely or none of the time”) to 4 (“most or all the time”). All items point in the same direction and a higher total CES-D score indicates a higher level of depression [31]. The CES-D was demonstrated to have acceptable to satisfactory psychometric properties [32]. For example, regarding internal consistency, the Cronbach α value derived for the CES-D was 0.84 [32]. Moreover, the Chinese version of the CES-D has been reported to exhibit favorable psychometric properties [33,34]. The Cronbach α value derived for the CES-D in the present study was 0.82.

#### 2.2.3. Suicidality Module of the Kiddie Schedule for Affective Disorders and Schizophrenia–Epidemiological Version

We adopted a six-item questionnaire based on the Kiddie Schedule for Affective Disorders and Schizophrenia–Epidemiological Version (K-SADS-E) [35] to assess the frequency of suicidal ideation and the number of suicide attempts in the preceding year [36]. All items of this questionnaire are “yes” or “no” questions. The total number of questions eliciting a “yes” response indicates the severity of suicide risk. The Cronbach α value derived for the suicidality module of the K-SADS-E in the present study was 0.70.

#### 2.2.4. Family and Peer Adaptation, Partnership, Growth, Affection, and Resolve Index

We used the five-item Chinese version [37] of the family and peer Adaptation, Partnership, Growth, Affection, and Resolve (APGAR) index [38] to assess five components of family and peer support: adaptability, partnership, growth, affection, and resolve. The items (e.g., “I am satisfied with the help that I receive from my family/peers when something is troubling me”) are rated on a 4-point Likert-type scale with anchors ranging from 1 (“never”) to 4 (“always”). A higher total score indicates a higher level of perceived family and peer support. The Cronbach α values derived for the family and peer APGAR index in the present study were 0.87 and 0.93, respectively.

#### 2.2.5. Positive and Negative Syndrome Scale

We used the Chinese version of the positive and negative modules of the Positive and Negative Syndrome Scale to assess the severity of the participants’ current positive and negative symptoms [39]. Each module contains seven items, which are rated on a 7-point Likert-type scale with anchors ranging from 1 (“absent”) to 7 (“extreme”). A higher mean score indicates a higher level of psychiatric symptoms.

### 2.3. Data Analysis

We analyzed the measure scores of the adopted scales (i.e., the three versions of UCLA-LSV3 at both baseline and retest, CES-D, suicidality module, family APGAR, and peer APGAR) as well as the participants’ characteristics by using descriptive statistics. We also used descriptive statistics to analyze the scores of the 20 UCLA-LSV3 items and demonstrate their properties. Confirmatory factor analysis (CFA), concurrent validity, internal consistency, and test–retest reliability assessments were performed to evaluate the psychometric properties of the three versions of UCLA-LSV3. Moreover, we performed CFA using the root mean square error of approximation (RMSEA) to determine the optimal sample size for our study [40]. We set the type I error to 0.05, desired power to 0.95, null RMSEA to 0, alternative RMSEA to 0.08, and degrees of freedom to 20; thus, we determined that the required sample size was 240.625. Next, we repeated the analysis using the same settings but changing the degrees of freedom to 170, resulting in a required sample size of 66. Because CFA requires a minimum of 200 participants for precise estimation [41], the present study recruited 300 participants to ensure strong and precise estimates.

For the CFA, an estimator based on diagonally weighted least squares was used to examine whether the present data fit the proposed unidimensional structure for all UCLA-LSV3 versions. The following fit indices were considered to indicate an acceptable data–model fit: comparative fit index (CFI) > 0.9, Tucker–Lewis index (TLI) > 0.9, RMSEA < 0.08, and standardized root mean square residual (SRMR) < 0.08 [42]. Pearson correlations were used to measure concurrent validity and test–retest reliability. Specifically, four external criterion measures (CES-D, suicidality module, family APGAR, and peer APGAR) were used to measure concurrent validity because previous studies have demonstrated associations between psychological distress, perceived support, and loneliness [8,16]. Additionally, we expected moderate to large effects (i.e., *r* > 0.3) [43] and large effects (i.e., *r* > 0.5) [40] on concurrent validity and test–retest reliability, respectively. For internal consistency, we adopted a Cronbach α value of >0.7 as an acceptable value [44].

We used R software (R Foundation for Statistical Computing, Vienna, Austria) with the *lavaan* package [45] to conduct the CFA and SPSS 20.0 (IBM, Armonk, NY, USA) to conduct all other analyses.

## 3. Results

Our study sample comprised 267 (89.0%) participants with schizophrenia and 33 (11.0%) participants with schizoaffective disorder. The mean age of the participants was 45.88 (standard deviation [SD] = 11.67) years, and the mean years of education was 13.01 (SD = 2.57) years. Slightly more than half of the participants were women (*n* = 161; 53.7%; mean age = 45.58 years; age range = 20–69 years). The mean duration of illness was 18.94 (SD = 10.17) years, indicating a chronic course of illness. The mean severity scores of positive and negative symptoms were 3.46 and 3.57, respectively, indicating mild to moderate symptoms. The participants’ characteristics, along with these scores, are presented in Table 1.

We observed that the scores of the 20 items of UCLA-LSV3 were normally distributed (skewness = from −0.01 to 0.79; kurtosis = from −1.15 to 0.06). Moreover, the scores of the 20 items of UCLA-LSV3, which were rated on Likert-type scales, ranged between 1.82 and 2.50 (Table 2). All three versions of UCLA-LSV3 were demonstrated to exhibit unidimensional structures (CFI = 0.95–1.00; TLI = 0.94–1.00; RMSEA = 0.000–0.072; SRMR = 0.000–0.055), but the SRMR value derived for the full version of UCLA-LSV3 was slightly high (0.089). Nevertheless, the perfect fit indices observed for the three-item version of UCLA-LSV3 should be interpreted with caution because the three-item structure was determined to be saturated in the CFA equation (Table 3).

Our analyses generally supported the concurrent validity of UCLA-LSV3 (CES-D: *r* = from 0.60 to 0.67, *p* < 0.001; family APGAR: *r* = from −0.48 to −0.32, *p* < 0.001; peer APGAR: *r* = from −0.25 to −0.51, *p* < 0.001; and suicidality module: *r* = 0.14–0.19, *p* = from 0.001 to 0.02). The strongest associations between the external criterion measures (CES-D, suicidality module, family APGAR, and peer APGAR) and UCLA-LSV3 were observed in the full version, and the weakest associations were observed in the three-item version (Table 4). The correlations between the full version of UCLA-LSV3 and the eight-item version, between the full version and the three-item version, and between the eight-item version and the three-item version were 0.92 (*p* < 0.001), 0.77 (*p* < 0.001), and 0.85 (*p* < 0.001), respectively. Moreover, UCLA-LSV3 demonstrated acceptable internal consistency and test–retest reliability (internal consistency: α = 0.90 for the full version, 0.75 for the eight-item version, and 0.71 for the three-item version; reliability: *r* = 0.86 for the full version, 0.84 for the eight-item version, and 0.7 for the three-item version).

## 4. Discussion

This study examined the psychometric properties of three versions of UCLA-LSV3 in 300 individuals with schizophrenia or schizoaffective disorder. In general, all three versions of UCLA-LSV3 exhibited acceptable to satisfactory psychometric properties in terms of unidimensional structures, concurrent validity with different external measures (i.e., CES-D, suicidality module, and family and peer APGAR), internal consistency, and test–retest reliability. The CFA fit indices confirmed the unidimensional structures of all three versions of UCLA-LSV3; however, the three-item version was determined to be saturated in the CFA model. Although all three versions of UCLS-LSV3 demonstrated acceptable to satisfactory concurrent validity, internal consistency, and test–retest reliability, the full version exhibited the best performance, followed by the eight-item version and then the three-item version. Moreover, the three versions of UCLA-LSV3 shared relatively strong associations with each other.

Our findings corroborate existing evidence regarding the factor structure of UCLA-LSV3; that is, UCLA-LSV3 was verified to exhibit a unidimensional structure that reflects the concept of loneliness [22]. According to our review of the literature, studies have provided psychometric evidence supporting the unidimensional structure of UCLA-LSV3 in various populations [8,9,10,11,12] but not in populations with schizophrenia or schizoaffective disorder. Accordingly, the present findings extend the psychometric evidence concerning the unidimensional structure of UCLA-LSV3 to a population with schizophrenia or schizoaffective disorder. The full version of UCLA-LSV3 and two short versions (the eight-item and three-item versions) were found to exhibit unidimensional structures, which are consistent with the concept of loneliness [22] and support existing evidence regarding the constructs of the two short versions [16,17]. However, the unidimensional structure of the three-item version of UCLA-LSV3 could be attributed to the saturation of our CFA model, a mathematical phenomenon that could lead to a consistently perfect model fit [46]. Nevertheless, our findings, as well as those of a study conducted in Hong Kong [16], reveal strong factor loadings for the three-item version of UCLA-LSV3. Therefore, the three-item version of UCLA-LSV3 can be considered to exhibit a unidimensional construct.

Several studies have suggested that UCLA-LSV3 has a multidimensional structure [23,24,25,26], and this proposal could be attributed to method effects (i.e., confounding effects caused by the use of different wording patterns) [8]. To test this proposal, empirical studies have examined the unidimensional structure of UCLA-LSV3 by conducting CFA adjusted for wording effects, and their findings fully support the unidimensional concept of loneliness and method effects [8,9,10]. Although the present study did not control for wording effects, the CFA results still support the unidimensional concept of loneliness. Therefore, we demonstrate that the full version, eight-item version, and three-item version of UCLA-LSV3 assess loneliness as a unidimensional construct.

We also compared the three versions of UCLA-LSV3 in terms of their construct validity as well as their attributes. All three versions were observed to have acceptable concurrent validity, internal consistency, and test–retest reliability in our participants with schizophrenia or schizoaffective disorder. Moreover, the full version of UCLA-LSV3 had the best performance, followed by the eight-item version and then the three-item version. The superior performance of the full version may be partially explained by the features of some psychometric testing statistics (e.g., Cronbach α), which are associated with the number of tested items (i.e., a measure with more items tends to have a more favorable Cronbach α value) [47]. Because the full version of UCLA-LSV3 outperformed the two short versions in all psychometric evaluations (except for the CFA), it could be the best choice for assessing loneliness in individuals with schizophrenia or schizoaffective disorder. Nevertheless, clinicians should select the version of the UCLA-LSV3 that best suits their clinical settings when assessing loneliness in individuals with schizophrenia or schizoaffective disorder. That is, time constraints and practical settings are the major factors that should be used to determine which version of UCLA-LSV3 to use. We recommend using the eight-item version of UCLA-LSV3 to assess individuals with schizophrenia or schizoaffective disorder in busy clinical settings in which time is restricted; however, whenever possible, clinicians should use the full version of the UCLA-LSV3 because it provides a more comprehensive and informative assessment than either short version.

We observed several associations between UCLA-LSV3 and the external criterion measures (CES-D, suicidality module, family APGAR, and peer APGAR); these associations can be explained by existing evidence regarding the relationships between loneliness, depression, suicidal risk, and perceived support [48,49,50,51]. A meta-analysis of 88 pooled reported that loneliness, a negative emotion, triggers subsequent depression with a moderate effect size [50]. Moreover, studies including individuals with schizophrenia or schizoaffective disorder have reported a moderate association between UCLA-LSV3 and the CES-D (*r* = 0.46) [2,48,50]. Therefore, the associations between UCLA-LSV3 and the CES-D noted in the present study (*r* = 0.60 to 0.67) corroborate previous observations regarding the association between loneliness and depression [2,48,50]. Similar to the mechanism underlying the association between loneliness and depression [50], loneliness could be an initial negative emotion that is subsequently associated with suicidal risk. Evidence demonstrates the association between loneliness and suicidal ideation among individuals with schizophrenia or schizoaffective disorder [48]. However, the association between loneliness and suicidal risk was weaker than that between loneliness and depression because the suicidal risk is a more severe condition than depression [51]. The association between UCLA-LSV3 and family and peer APGAR can be explained by the effects of perceived support on loneliness. Specifically, when an individual perceives less support from peers or family, the individual is likely to feel lonely and develop loneliness [52]. A previous meta-analysis reported a moderate level of negative association between loneliness and perceived support (*r* = from −0.48 to −0.34) [52], which is comparable to the present study’s findings (*r* = from −0.51 to −0.25).

Our findings have some clinical implications. First, the 20-item, 8-item, and 3-item versions of UCLA-LSV3 were all revealed to be effective instruments for assessing loneliness in individuals with schizophrenia or schizoaffective disorder. However, the full version of UCLA-LSV3 provides more comprehensive information and correlates more strongly with associated concepts (i.e., depression, suicidal risk, and perceived support) than the two shorter versions. Accordingly, clinicians should use the full version of UCLA-LSV3 whenever possible. Second, when using UCLA-LSV3 to assess loneliness in individuals with schizophrenia or schizoaffective disorder, the assessors or mental health professionals need not control for different structures of loneliness or stratify loneliness into different concepts; this is because UCLA-LSV3 provides only one score for the concept of loneliness. Third, our findings confirm the unidimensional structure of UCLA-LSV3; therefore, mental health professionals can use the total UCLA-LSV3 score to evaluate loneliness and related factors in individuals with schizophrenia or schizoaffective disorder.

Some limitations in the present study should be addressed. First, UCLA-LSV3, the CES-D, the suicidality module of the K-SADS-E, and the family and peer APGAR were self-administered by the participants; therefore, single-rater biases may have existed. Second, the participants were recruited from psychiatric outpatient units. Therefore, the applicability of UCLA-LSV3 to other groups of individuals with schizophrenia or schizoaffective disorder (e.g., those living in chronic psychiatric wards or those who do not visit psychiatric medical units for treatment) is still unknown. Third, some key psychometric properties (e.g., responsiveness) were not examined in this study. Accordingly, future studies should examine whether all three versions of UCLA-LSV3 have satisfactory responsivity.

## 5. Conclusions

In conclusion, all three versions of UCLA-LSV3 tested in the present study exhibited acceptable to satisfactory psychometric properties in individuals with schizophrenia. Specifically, UCLA-LSV3 exhibited a unidimensional structure, acceptable internal consistency, satisfactory test–retest reliability, and high concurrent validity for depression, suicidal risk, and perceived support. Additionally, the 20-item version of UCLA-LSV3 was demonstrated to have superior psychometric performance compared with the shorter versions; therefore, we recommend its use whenever possible. Nevertheless, when deliberating which version of UCLA-LSV3 is the best choice for assessing loneliness in individuals with schizophrenia or schizoaffective disorder, healthcare providers and therapists should consider time availability and practicality.

## Figures and Tables

**Table 1 ijerph-19-08443-t001:** Participant characteristics (*n* = 300).

Variable	M (SD) or *n* (%)
Age (year)	45.88 (11.67)
Sex	
Men	139 (46.3)
Women	161 (53.7)
Diagnosis	
Schizophrenia	267 (89.0)
Schizoaffective disorder	33 (11.0)
Years of education (year)	13.01 (2.57)
Duration of illness (year)	18.94 (10.17)
Positive symptoms	3.46 (0.88)
Negative symptoms	3.57 (0.93)
UCLA Loneliness Scale (Version 3)	
Baseline full version score	2.17 (0.55)
Retest full version score	2.02 (0.49)
Baseline 8-item version score	2.24 (0.58)
Retest 8-item version	2.03 (0.52)
Baseline 3-item version score	2.17 (0.78)
Retest 3-item version score	1.89 (0.59)
CES-D score	16.56 (10.82)
Suicidal risk score	0.42 (0.91)
Family APGAR score	15.67 (3.64)
Friend APGAR score	13.31 (4.40)

Abbreviations: CES-D—Center for Epidemiological Studies Depression Scale; APGAR—Adaptation, Partnership, Growth, Affection, and Resolve. Note: the retest of the UCLA Loneliness Scale (Version 3) was completed by 50 participants 1 month after the original test.

**Table 2 ijerph-19-08443-t002:** Score distributions of the 20 items comprising the UCLA Loneliness Scale, Version 3 (*n* = 300).

Item	M (SD)	*n* (%)				Skewness	Kurtosis
		Score 1	Score 2	Score 3	Score 4		
Item 1 ^a^	1.82 (0.82)	120 (40.0)	126 (42.0)	42 (14.0)	12 (4.0)	0.79	0.06
Item 2	2.30 (1.01)	82 (27.3)	86 (28.7)	93 (31.0)	39 (13.0)	0.14	−1.11
Item 3	2.08 (0.95)	97 (32.3)	107 (35.7)	70 (23.3)	26 (8.7)	0.45	−0.78
Item 4	2.19 (1.01)	98 (32.7)	79 (26.3)	91 (30.3)	32 (10.7)	0.24	−1.15
Item 5 ^a^	2.18 (0.99)	86 (28.7)	112 (37.3)	63 (21.0)	39 (13.0)	0.43	−0.85
Item 6 ^a^	2.44 (0.96)	51 (17.0)	116 (38.7)	84 (28.0)	49 (16.3)	0.16	−0.90
Item 7	1.92 (0.89)	113 (37.7)	114 (38.0)	56 (18.7)	17 (5.7)	0.64	−0.42
Item 8	2.28 (0.99)	79 (26.3)	94 (31.3)	91 (30.3)	36 (12.0)	0.17	−1.03
Item 9 ^a^	2.11 (0.91)	86 (28.7)	120 (40.0)	70 (23.3)	24 (8.0)	0.43	−0.65
Item 10 ^a^	2.31 (0.90)	60 (20.0)	116 (38.7)	95 (31.7)	29 (9.7)	0.15	−0.76
Item 11	2.09 (0.95)	96 (32.0)	108 (36.0)	69 (23.0)	27 (9.0)	0.45	−0.77
Item 12	1.82 (0.88)	134 (44.7)	100 (33.3)	52 (17.3)	14 (4.7)	0.77	−0.33
Item 13	2.11 (0.97)	100 (33.3)	92 (30.7)	82 (27.3)	26 (8.7)	0.34	−0.99
Item 14	2.12 (0.97)	101 (33.7)	89 (29.7)	82 (27.3)	28 (9.3)	0.34	−1.02
Item 15 ^a^	2.31 (0.97)	70 (23.3)	108 (36.0)	82 (27.3)	40 (13.3)	0.23	−0.94
Item 16 ^a^	2.26 (0.94)	74 (24.7)	104 (34.7)	92 (30.7)	30 (10.0)	0.18	−0.92
Item 17	2.38 (1.00)	71 (23.7)	87 (29.0)	100 (33.3)	42 (14.0)	0.05	−1.07
Item 18	2.50 (0.98)	54 (18.0)	95 (31.7)	98 (32.7)	53 (17.7)	−0.01	−1.01
Item 19 ^a^	2.13 (0.91)	83 (27.7)	120 (40.0)	72 (24.0)	25 (8.3)	0.40	−0.68
Item 20 ^a^	2.00 (0.87)	94 (31.3)	130 (43.3)	58 (19.3)	18 (6.0)	0.56	−0.36

^a^ These items were reverse coded.

**Table 3 ijerph-19-08443-t003:** Factor loading and fit indices in the confirmatory factor analyses of three versions of the UCLA Loneliness Scale (Version 3) (*n* = 300).

Factor Loading	Full Version	8-Item Version	3-Item Version
Item 1: How often do you feel that you are “in tune” with the people around you?	0.57	--	--
Item 2: How often do you feel that you lack companionship?	0.47	0.55	0.54
Item 3: How often do you feel that there is no one you can turn to?	0.57	0.58	--
Item 4: How often do you feel alone?	0.65	--	--
Item 5: How often do you feel that you are part of a group of friends?	0.58	--	--
Item 6: How often do you feel that you have a lot in common with the people around you?	0.39	--	--
Item 7: How often do you feel that you are no longer close to anyone?	0.61	--	--
Item 8: How often do you feel that your interests and ideas are not shared by those around you?	0.56	--	--
Item 9: How often do you feel outgoing and friendly?	0.42	0.26	--
Item 10: How often do you feel close to people?	0.61	--	--
Item 11: How often do you feel left out?	0.62	0.64	0.69
Item 12: How often do you feel that your relationships with other people are not meaningful?	0.66	--	--
Item 13: How often do you feel that no one really knows you well?	0.73	--	--
Item 14: How often do you feel isolated from others?	0.69	0.75	0.79
Item 15: How often do you feel that you can find companionship when you want it?	0.55	0.39	--
Item 16: How often do you feel that there are people who really understand you?	0.59	--	--
Item 17: How often do you feel shy?	0.32	0.34	--
Item 18: How often do you feel that people are around you but not with you?	0.52	0.54	--
Item 19: How often do you feel that there are people you can talk to?	0.54	--	--
Item 20: How often do you feel that there are people you can turn to?	0.52	--	--
**Fit statistics**			
χ^2^ (df)	433.14 (170)	30.04 (20)	0 (0) ^a^
*p*-value	<0.001	<0.001	-- ^a^
CFI	0.95	0.98	1.00 ^a^
TLI	0.94	0.98	1.00 ^a^
RMSEA	0.072	0.041	0.000 ^a^
90% CI of RMSEA	0.064, 0.080	0.000, 0.069	0.000, 0.000 ^a^
SRMR	0.089	0.055	0.000 ^a^

^a^ Perfect fit occurs because this model only contains three items, which is a saturated model in the confirmatory factor analysis equation. Abbreviations: CFI—comparative fit index; TLI—Tucker–Lewis index; RMSEA—root mean square error of approximation; SRMR—standardized root mean square residual.

**Table 4 ijerph-19-08443-t004:** Concurrent validity, test–retest reliability, and internal consistency of the three versions of the UCLA Loneliness Scale, Version 3 (*n* = 300).

	Full Version	8-Item Version	3-Item Version
Full version (α = 0.90)	test–retest = 0.86	--	--
8-item version (α = 0.75)	0.92	test–retest = 0.84	--
3-item version (α = 0.71)	0.77	0.85	test–retest = 0.77
CES-D	0.67	0.65	0.60
Suicidal risk	0.19 (*p* = 0.001)	0.18 (*p* = 0.002)	0.14 (*p* = 0.02)
Family APGAR	−0.48	−0.39	−0.32
Friend APGAR	−0.51	−0.37	−0.25

Abbreviations: CES-D—Center for Epidemiological Studies Depression; APGAR—Adaptation, Partnership, Growth, Affection, and Resolve. Note: all *p* values are <0.001, except for those specifically mentioned; the retest of the UCLA Loneliness Scale (Version 3) was completed by 50 participants 1 month after the original test.

## Data Availability

The data will be available upon reasonable request to the corresponding authors.

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
