# Peer review of "Comparing the Psychometric Properties among Three Versions of the UCLA Loneliness Scale in Individuals with Schizophrenia or Schizoaffective Disorder"

_ijerph, 2022, doi:10.3390/ijerph19148443_

Round 1
Reviewer 1 Report
Please explain how to perform data collection in detail?
Please explain which sample method used in this study?
Please explain how to calculate sample size?
Please, use the current references in discussion section. This section is short, must be improved more..
Please add practical applications after or before limitations..
Author Response
We appreciated your valuable comment. As discussed below, we have revised our manuscript with underlines based on your suggestions. Please let us know if we need to provide anything else regarding this revision.
Comment
Please explain how to perform data collection in detail?
Response
We added explanation for perform data collection in this study. Please refer to line 137-143.
“Trained research assistants conducted face-to-face interviews with the participants in the interview rooms of the psychiatric outpatient unit affiliated with Kaohsiung Medical University Hospital. The interviews were conducted to collect the participants’ responses regarding experiences of loneliness, depression, suicidal ideation, and perceived support from family and friends. The participants were assured that their responses would remain confidential. Each interview lasted 25 to 40 minutes, varying by the participant.”
Comment
Please explain which sample method used in this study?
Response
This study used convenience sampling to recruit participants. We the added sample method into line 127.
“The present study used convenience sampling…”
Comment
Please explain how to calculate sample size?
Response
Response: We have now provided the information regarding how we decided the sample size in the revised manuscript. Please refer to line 198-206.
“Moreover, we performed CFA using the root mean square error of approximation (RMSEA) to determine the optimal sample size for our study [40]. We set the type I error to 0.05, desired power to 0.95, null RMSEA to 0, alternative RMSEA to 0.08, and degrees of freedom to 20; thus, we determined that the required sample size was 240.625. Next, we repeated the analysis using the same settings, but changing the degrees of freedom to 170, resulting in a required sample size of 66. Because CFA requires a minimum of 200 participants for precise estimation [41], the present study recruited 300 participants to ensure strong and precise estimates.”
Comment
Please, use the current references in discussion section. This section is short, must be improved more.
Response
Thank you for the suggestion. We have now elaborated more in the Discussion section. Specifically, we have added a paragraph to describe why loneliness is associated with depression, suicidal risk, and perceived support. Please refer to line 332-353.
“We observed several associations between UCLA-LSV3 and the external criterion measures (CES-D, suicidality module, family APGAR, and peer APGAR); these associations can be explained by existing evidence regarding the relationships between loneliness, depression, suicidal risk, and perceived support [48-51]. A meta-analysis of 88 pooled reported that loneliness, a negative emotion, triggers subsequent depression with a moderate effect size [50]. Moreover, studies including individuals with schizophrenia or schizoaffective disorder have reported a moderate association between UCLA-LSV3 and the CES-D (r = 0.46) [48,50,51]. Therefore, the associations between UCLA-LSV3 and the CES-D noted in the present study (r = 0.60 to 0.67) corroborate previous observations regarding the association between loneliness and depression [48,50,51]. Similar to the mechanism underlying the association between loneliness and depression [50], loneliness could be an initial negative emotion that is subsequently associated with suicidal risk. Evidence demonstrates the association between loneliness and suicidal ideation among individuals with schizophrenia or schizoaffective disorder [48]. However, the association between loneliness and suicidal risk was weaker than that between loneliness and depression because suicidal risk is a more severe condition than depression [52]. The association between UCLA-LSV3 and family and peer APGAR can be explained by the effects of perceived support on loneliness. Specifically, when an individual perceives less support from peers or family, the individual is likely to feel lonely and to develop loneliness [53]. A previous meta-analysis reported a moderate level of negative association between loneliness and perceived support (r = −0.48 to −0.34) [53], which is comparable to the present study’s findings (r = −0.51 to −0.25).”
Comment
Please add practical applications after or before limitations.
Response
We have now added the practical applications before the Limitations section. Please refer to line 354-366.
“Our findings have some clinical implications. First, the 20-item, 8-item, and 3-item versions of UCLA-LSV3 were all revealed to be effective instruments for assessing loneliness in individuals with schizophrenia or schizoaffective disorder. However, the full version of UCLA-LSV3 provides more comprehensive information and correlates more strongly with associated concepts (i.e., depression, suicidal risk, and perceived support) than do the two shorter versions. Accordingly, clinicians should use the full version of UCLA-LSV3 whenever possible. Second, when using UCLA-LSV3 to assess loneliness in individuals with schizophrenia or schizoaffective disorder, the assessors or mental health professionals need not control for different structures of loneliness or stratify loneliness into different concepts; this is because UCLA-LSV3 provides only one score for the concept of loneliness. Third, our findings confirm the unidimensional structure of UCLA-LSV3; therefore, mental health professionals can use the total UCLA-LSV3 score to evaluate loneliness and related factors in individuals with schizophrenia or schizoaffective disorder.”
Reviewer 2 Report
The present study reported to examine the unidimensional structure, internal consistency, concurrent validity, and test-retest reliability of the Chinese version of the UCLA-LSV3 across the full version with 20 items, the 8-item version, and the 3-item version. The current study is on a topic of relevance and general interest to the readers of the journal. On the one hand, I found the paper overall well written and well described. Therefore, I recommend a major revision.
1. It is recommended to modify the title to be more attractive.
2. In the abstract section; this part should be carefully reviewed, highlighting the novelty of the work, the main objective and the main results obtained.
3. In the introduction part: The following points should be taken into consideration:
a- To highlight the importance of the current work, the introductory part should be reviewed, highlighting the main problem with brief information, the existing challenges, the available solutions, and the importance of the work.
b- In addition to highlighting the advantages of the current study.
4. The whole English should improve.
5. Conclusion needs to be rewritten
Author Response
We appreciated your valuable comment. As discussed below, we have revised our manuscript with underlines based on your suggestions. Please let us know if we need to provide anything else regarding this revision.
Comment
It is recommended to modify the title to be more attractive.
Response
Thank you for your suggestion we modified the title into “Comparing the Psychometric Properties among Three Versions of the UCLA Loneliness Scale in Individuals with Schizophrenia or Schizoaffective Disorder.” Please refer to line 2-4.
Comment
In the abstract section; this part should be carefully reviewed, highlighting the novelty of the work, the main objective and the main results obtained.
Response
Thank you for your suggestion. We revised the contents of Abstract below.
Novelty of the work: “…the validity of this scale for assessing loneliness in individuals with schizophrenia or schizoaffective disorder has not been determined. Additionally, studies validating the eight-item and three-item versions of UCLA-LSV3 have not included individuals with severe mental illness; therefore, whether the short versions are comparable to the full 20-item version of UCLA-LSV3 for this population is unclear.” Please refer to line 36-40.
The main objective: “The present study examined the unidimensional structure, internal consistency, concurrent validity, and test–retest reliability of the Chinese versions of UCLA-LSV3 (i.e., 20-item, 8-item, and 3-item versions) to determine which version is most appropriate for assessing loneliness in individuals with schizophrenia or schizoaffective disorder in Taiwan.” Please refer to line 40-44.
The main results obtained: “The results indicated that all three versions of UCLA-LSV3 exhibited acceptable to satisfactory psychometric properties in terms of unidimensional constructs, concurrent validity, and test–retest reliability. The full version of UCLA-LSV3 had the best performance, followed by the eight-item version and the three-item version. Moreover, the three versions had relatively strong associations with each other. Therefore, when deliberating which version of UCLA-LSV3 is the best choice for assessing loneliness in individuals with schizophrenia or schizoaffective disorder, health-care providers and therapists should consider time availability and practicality.” Please refer to line 51-57.
Comment
In the introduction part: The following points should be taken into consideration: To highlight the importance of the current work, the introductory part should be reviewed, highlighting the main problem with brief information, the existing challenges, the available solutions, and the importance of the work.
Response
Thank you for your suggestion. We revised Introduction section of the manuscript based on your suggestion.
- Main problem and existing challenges: Please refer to line 77-109.
“Despite the validity and reliability of UCLA-LSV3, the psychometric properties of this scale and its other versions warrant attention. First, the Chinese versions of UCLA-LSV3 have never been validated among individuals with schizophrenia or schizoaffective disorder in Taiwan, according to a review of the literature; therefore, further psychometric evidence must be derived for these versions.
Second, different versions of UCLA-LSV3 have been proposed, but studies have not compared their psychometric properties or evidence regarding their suitability for assessing individuals with schizophrenia or schizoaffective disorder. In addition to the full version of UCLA-LSV3, which contains 20 items, several short versions of UCLA-LSV3 have been developed [16,18,19]. Among them, the eight- and three-item versions of UCLA-LSV3 have been recommended because their psychometric properties have been determined to be stronger than those of other short versions [18,20]. The eight- and three-item Chinese versions of UCLA-LSV3 have been tested for their psychometric properties, which have been reported to be satisfactory [16,17]. Nevertheless, the eight- and three-item versions of UCLA-LSV3 have not been tested on individuals with severe mental illnesses; therefore, the psychometric properties of these versions for populations with schizophrenia or schizoaffective disorder are unknown. Schizophrenia or schizoaffective disorder may affect an individual’s cognition [21]; thus, individuals with these disorders are likely to interpret UCLA-LSV3 differently when compared with individuals without said disorders. Accordingly, mental health professionals must be provided with information explaining how different versions of UCLA-LSV3 (the full 20-item version, 8-item version, and 3-item version) perform when applied to individuals with schizophrenia or schizoaffective disorder.
Third, the factor structure of UCLA-LSV3 is still uncertain. By definition, loneliness should be considered a unidimensional concept [22]; however, empirical evidence regarding the psychometric properties of UCLA-LSV3 and other instruments for assessing loneliness reveals that these instruments may be multidimensional [23-29]. Nevertheless, studies have demonstrated a clear pattern of the unidimensionality of loneliness when controlling for method effects (i.e., positively and negatively worded items) in global versions of UCLA-LSV3, namely the Farsi [9], Turkish [10], and English versions [8]. Accordingly, the 20-, 8-, and 3-item versions of UCLA-LSV3 should all be treated as unidimensional instruments for assessing loneliness, although their factor structures could be influenced by method effects.”
- Available solutions, the importance and advantages of the current study: Please refer to line 113-124.
“...we tested and compared these three versions in terms of internal consistency, concurrent validity, and test–retest reliability to determine which version is most appropriate for assessing loneliness in individuals with schizophrenia or schizoaffective disorder. We observed that shorter versions of UCLA-LSV3 may reduce the time burden for obtaining loneliness information, whereas longer versions may provide comprehensive information of loneliness. Accordingly, mental health professionals must know how much information could be lost when they use a shorter version and must be able to make informed decisions about whether the 20-item version is worth the additional time investment. Moreover, if the scenarios for the shorter versions of UCLA-LSV3 are comparable to those for the full 20-item version, mental health professionals can simply use the shorter versions to reduce the administrative burden of completing the 20-item version.”
Comment
The whole English should improve.
Response
We invited another English-native editor to improve the English. Please refer to the certificate for English editing.
Comment
Conclusion needs to be rewritten
Response
We have now rewritten the Conclusion section. Please refer to line 378-387.
“In conclusion, all three versions of UCLA-LSV3 tested in the present study exhibited acceptable to satisfactory psychometric properties in individuals with schizophrenia. Specifically, UCLA-LSV3 exhibited a unidimensional structure, acceptable internal consistency, satisfactory test–retest reliability, and high concurrent validity for depression, suicidal risk, and perceived support. Additionally, the 20-item version of UCLA-LSV3 was demonstrated to have superior psychometric performance compared with the shorter versions; therefore, we recommend its use whenever possible. Nevertheless, when deliberating which version of UCLA-LSV3 is the best choice for assessing loneliness in individuals with schizophrenia or schizoaffective disorder, health-care providers and therapists should consider time availability and practicality.”
Reviewer 3 Report
The article is interesting and provides additional information on the subjective feeling of loneliness in patients with schizophrenia.
The article requires the following corrections - specifying the age of women - before menopause and after menopause – hormonal effects may be additional important in the feeling of loneliness. The disease state of the patients is not specified. 3 types of scales should be included in the tables, as it is not clear how the questions on the individual scales are formulated.
Does the comparison of the three types of scales affect the APGAR scale, as such conclusions can be drawn from these results?
Please explain why the 20 items scale was compared to the 8 and 3 items scales?
Author Response
We appreciated your valuable comment. As discussed below, we have revised our manuscript with underlines based on your suggestions. Please let us know if we need to provide anything else regarding this revision.
Comment
Specifying the age of women - before menopause and after menopause – hormonal effects may be additional important in the feeling of loneliness.
Response
Thank you for your comment. We added the age of women into the revised manuscript (mean age = 45.58 years; age range = 20–69 years). Please refer to line 227. We agreed that hormonal effects after menopause may be additional important in the feeling of loneliness for women, although we did not survey the menstrual status of female participants or the cause of menopause (age or antipsychotic induced) in this study. We will examine the factors related to the level of loneliness in patient with schizophrenia or schizoaffective disorder in another manuscript. We added this description into the revised manuscript (“…our findings confirm the unidimensional structure of UCLA-LSV3; therefore, mental health professionals can use the total UCLA-LSV3 score to evaluate loneliness and related factors in individuals with schizophrenia or schizoaffective disorder.”) Please refer to line 364-366.
Comment
The disease state of the patients is not specified.
Response
Thank you for your comment. We added the duration of illness and severities of positive and negative symptoms into the revised manuscript. Please refer to line 185-190, line 227-230, and Table 1.
“2.2.5. Positive and Negative Syndrome Scale
We used the Chinese version of the positive and negative modules of the Positive and Negative Syndrome Scale to assess the severity of the participants’ current positive and negative symptoms [39]. Each module contains seven items, which are rated on a 7-point Likert-type scale with anchors ranging from 1 (“absent”) to 7 (“extreme”). A higher mean score indicates a higher level of psychiatric symptoms.”
“The mean duration of illness was 18.94 (SD = 10.17) years, indicating a chronic course of illness. The mean severity scores of positive and negative symptoms were 3.46 and 3.57, respectively, indicating mild to moderate symptoms.”
Comment
3 types of scales should be included in the tables, as it is not clear how the questions on the individual scales are formulated.
Response
We have now added the item contents on Table 3. We believe that readers will clearly know the item contents for each UCLA Loneliness Scale, Version 3 in the three forms. Please refer to line 249.
Comment
Does the comparison of the three types of scales affect the APGAR scale, as such conclusions can be drawn from these results?
Response
Thank you for raising this concern. We have now described that the relationships found between UCLA Loneliness Scale, Version 3 and the APGAR scale could be due to the reason that less perceived support leads to more loneliness. Please refer to line 347-353.
“The association between UCLA-LSV3 and family and peer APGAR can be explained by the effects of perceived support on loneliness. Specifically, when an individual perceives less support from peers or family, the individual is likely to feel lonely and to develop loneliness [53]. A previous meta-analysis reported a moderate level of negative association between loneliness and perceived support (r = −0.48 to −0.34) [53], which is comparable to the present study’s findings (r = −0.51 to −0.25).”
Comment
Please explain why the 20 items scale was compared to the 8 and 3 items scales?
Response
We have now made the explanation. Please refer to line 113-124.
“…we tested and compared these three versions in terms of internal consistency, concurrent validity, and test–retest reliability to determine which version is most appropriate for assessing loneliness in individuals with schizophrenia or schizoaffective disorder. We observed that shorter versions of UCLA-LSV3 may reduce the time burden for obtaining loneliness information, whereas longer versions may provide comprehensive information of loneliness. Accordingly, mental health professionals must know how much information could be lost when they use a shorter version and must be able to make informed decisions about whether the 20-item version is worth the additional time investment. Moreover, if the scenarios for the shorter versions of UCLA-LSV3 are comparable to those for the full 20-item version, mental health professionals can simply use the shorter versions to reduce the administrative burden of completing the 20-item version.”
Round 2
Reviewer 2 Report
The current revision meets the standards of the paper and it is recommended to accept the paper for publication.
Reviewer 3 Report
Article corrected as expected. The ambiguities have been properly cleared up.